pattern recognition/psychology

convolutional neural nets, automatic face recognition, human face matching

**Author for correspondence:**
Peter J. B. Hancock
e-mail: pjbh1@stir.ac.uk

# Convolutional neural net face recognition works in non-human-like ways

Peter J. B. Hancock, Rosyl S. Somai and Viktoria R. Mileva

Psychology, Faculty of Natural Sciences, University of Stirling, FK9 4LA, Scotland

PJBH, 0000-0001-6025-7068; RSS, 0000-0002-8644-1282; VRM, 0000-0002-7983-3069

Convolutional neural networks (CNNs) give the state-of-the-art performance in many pattern recognition problems but can be fooled by carefully crafted patterns of noise. We report that CNN face recognition systems also make surprising 'errors'. We tested six commercial face recognition CNNs and found that they outperform typical human participants on standard face-matching tasks. However, they also declare matches that humans would not, where one image from the pair has been transformed to appear a different sex or race. This is not due to poor performance; the best CNNs perform almost perfectly on the human face-matching tasks, but also declare the most matches for faces of a different apparent race or sex. Although differing on the salience of sex and race, humans and computer systems are not working in completely different ways. They tend to find the same pairs of images difficult, suggesting some agreement about the underlying similarity space.

## 1. Introduction

Convolutional neural networks (CNNs) have transformed pattern recognition, achieving the state-of-the-art performance in many applications, including automated face recognition (AFR) [1]. However, they can be deceived by noise patterns, either on their own or added to another image [2]. For example, an image that to humans looks like a dog might be classified as a penguin. The current controversies around the public use of AFR and lack of clear legislation have resulted in a ban on its use in some places. Therefore, it is important to investigate whether similarly surprising results apply to faces.

What can you say about the people depicted in figure 1? The second image figure 1*b* is a composite made of several male actors. The first figure 1*a* is the same face, transformed to look female, the third figure 1*c*, the same face transformed to look Black. This work originated with the observation that a state-of-the-art CNN face

**Figure 1.** Variations on a face: (*a*) transformed to look female, (*b*) the original, a composite of a number of male actors, (*c*) transformed to look Black.

recognition system reports that images (*a*) and (*c*) are both a match for (*b*). This is surprising, because to the human observer image (*a*) appears to be a different sex, and image (*c*) a different race to image (*b*).

For humans, the processing of characteristics of faces such as their apparent race and sex is thought to be obligatory; we do so whether or not it is relevant to the task in hand. Karnadewi & Lipp [3] used a Garner task to show that while changing the face to one of a different race, sex or age affected decisions about expression, the effect was asymmetric and changing expression had no effect on decisions about race, sex or age. Davidenko *et al*. [4] found that prompting observers to focus on the perceived race or sex of a face had no effect on the strength of adaptation, where, for example, looking at a male face makes a subsequent face look more female. Focusing on sex or race did not affect how much it was processed. Zhou *et al*. [5] use brain imaging methodology to show that 'other race' faces (White or Black to their Chinese participants) are identified early in the processing of a face image.

People categorize binary sex rapidly and with a high degree of accuracy [6]. A change of apparent sex affects decisions about identity. Campenella *et al*. [7] created morph continua between unfamiliar male and female faces, producing images that differed in equal-sized steps. Observers judged a pair of images that straddled the male–female boundary to differ more than a pair that did not. The apparent change in sex prompted a perceived change of identity.

To test the generality of our observation that a CNN does not share this bias and to try and understand something of the reason, we tested six commercial CNN face recognition engines during July and August 2019. In alphabetical order, these were Amazon Rekognition, Face++, FaceSoft, FaceX, Kairos and Microsoft. The testing reported here absolutely does not speak to whether one system is better than another. Rather, our aim was to examine whether this insensitivity to changes in apparent race or sex is common to a variety of different CNNs. In what follows therefore, these systems will be referred to only by a number, in a different order to the list above.

The systems were tested on two types of task. The first type was four matching tasks designed to be difficult for human observers. In these matching tasks, human participants are shown two images and must decide whether they show the same person or two different people. Some pairs are 'matched' (same identity), and some are 'mismatched' (different identity). The second set of tests involved matching tasks with faces that had been transformed to appear either a different sex or a different race, using PsychoMorph (figure 2) [8]. The CNNs were tested on the match between a transformed face and a different original photograph of the same person.

To be clear, we manipulated perceived sex and race only so that we could simulate a change in apparent identity for human observers. Other manipulations such as changing the weight or the application of facial hair would not necessarily have led human observers to perceive a change in identity. We used PsychoMorph in order to generate images that would otherwise match. Our study makes no assumptions of sex as a binary concept, nor does it speak to or have implications for transgender people. We do not know whether the changes observed in these perceived sex transforms are similar to the changes observed in individuals receiving hormonal treatment or surgery. Similarly, 'race' is not a well-defined concept; we use it here in the sense that people of different 'races' have different average facial appearance. Our use of a 'race transform' is a way to produce changes in a face that are perceived as a change of apparent identity and are not intended to imply or predict anything regarding features, characteristics, or stereotypes of Black people.

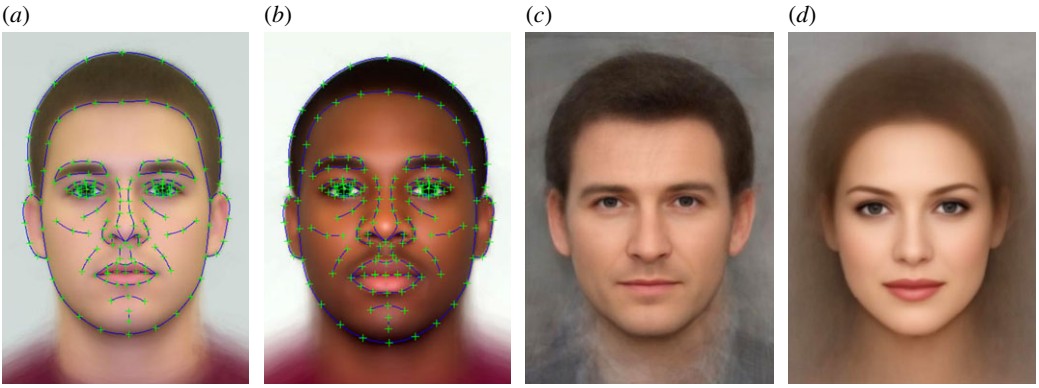

**Figure 2.** (*a,b*) White and Black male average images, showing the 179 reference points used. (*c,d*) average male and female faces. (*a* and *c*) appear different partly due to differences in camera to subject distance.

# 2. Materials and methods

## 2.1. Human matching tasks

We used (i) the Kent Face-Matching Task (Kent) [9], (ii) the Models Matching Task (Models) [10], (iii) the Make-up Task (Make-up) using pictures of women with and without heavy make-up that were obtained from YouTube videos [11] and (iv) the Dutch Matching Task (Dutch), using images of two Dutch TV personalities chosen for their similar appearance [12].

The long version of the Kent task [9] consists of 200 match trials and 20 mismatch trials constructed from a set of 252 pairs of colour images. In each pair, one is a studio picture showing head and shoulders at a resolution of 283 × 332 pixels, the other is a passport style image, taken some months earlier, showing just the head at a resolution of 142 × 192 pixels. The human data reported here come from the original paper [9]; there were 50 participants with an average age of 19.5 years.

The Models task uses pictures of male models who often vary in appearance markedly in different shoots [10] and consists of a total of 90 trials, half matched and half mismatched, in three blocks (*a–c*). The three sets were constructed to be of equal difficulty, based on pilot testing. All pairs were presented side by side, in colour, at a resolution of 300 × 420 pixels. The human data reported here came from 80 participants, average age 22, who had been tested on sets (*a*) and (*b*) only [13]. The computer systems were tested on all 90 trials.

The Make-up task, created in our laboratory, uses images of YouTube make-up videos posted by vloggers from non-English speaking countries, to reduce the chance of familiarity for our participants [11]. Four images of each vlogger were sourced, two with and two without make-up. Pairs of images consisted of both with make-up, both without make-up, and one with and one without make-up. Half of the trials were matched, and half were mismatched. Images were presented side by side, at a resolution of 320 pixels square, in colour and tightly cropped around the face. There were 294 pairs in total. We collected data from 48 participants, average age 26. Each was tested on 48 trials, 16 in each make-up condition, with the set used counterbalanced across participants to test 288 image pairs in total. The computer systems were tested on all 294 pairs.

The Dutch task, another in-house created test, consisted of 96 trials featuring two Dutch TV presenters, Bridget Maasland and Chantal Janzen. Half the trials were matched and half mismatched. Of the mismatch trials, half consisted of one image of each presenter, while the other half showed one of the presenters with a similar-looking third person (different for each pair). The images chosen were highly varied, including some that were tightly cropped around the face, and ranged in size from 187 × 184 pixels, to 548 × 500. We collected data from 60 participants, average age 22.

For all the sets, the mismatch items were chosen by the experimenters who created the task, based on visual similarity (e.g. same age, hair colour, eye colour).

## 2.2. Face transforms

The transformed images were generated using Psychomorph [8]. This can compute the average appearance of a set of face images, for example, White male actors (figure 2*a*). It then computes *the*

**Table 1.** Sex and race classifications returned by the Kairos system for the original and transformed Glasgow faces.

| | | female | male | White | Black | Hispanic | Asian | unclassified |
|---|---|---|---|---|---|---|---|---|
| female | original images | 124 | 8 | 122 | 1 | 2 | 7 | 0 |
| N = 132 | after race transform | 27 | 105 | 0 | 128 | 0 | 2 | 2 |
| | after sex transform | 0 | 132 | 124 | 0 | 0 | 8 | 0 |
| male | original images | 0 | 170 | 138 | 2 | 5 | 24 | 1 |
| N = 170 | after race transform | 0 | 170 | 0 | 165 | 0 | 3 | 2 |
| | after sex transform | 129 | 41 | 144 | 2 | 3 | 19 | 2 |

*difference*, in shape and colour, between this and another average, for example, an average of a set of images of White females or African American males and applies this *vector difference* to a third image, to change its apparent appearance. For instance, by adding the sex vector difference to a female identity, one can transform this identity to appear male while keeping all remaining information constant. That is, the new face deviates from the male average in the same way that the original deviated from the female average. This method was used to generate the images in figure 1. The CNNs were tested on pairs made up of a face transformed by apparent sex or race and a different, unaltered photograph of the same person. Before any transformation, all these pairs should therefore be declared as matched; the question is whether they are still declared a match when one of the faces appears to be a different sex or race.

The faces for transformation came from the Glasgow Unfamiliar Face database [14], sets C1 and DV (images of the same people, taken with different cameras), which we had previously 'marked up' for use in Psychomorph [8] using the 179 points shown in figure 2. Faces were transformed by apparent sex or race by adding the vector difference between the two average faces to each original face. In the case of the sex transform, 120% of the difference was added, to produce an image more clearly of the opposite sex in all cases. The C1 images were transformed, and the DV images used to test for a match.

The images used to perform the male race transform (figure 2*a,b*) came from averaging images of White and Black students collected by Chris Meissner at the University of South Florida, and for the female set from Lisa DeBruine, collected in London. The male and female averages (figure 2*c,d*) were produced at Stirling, using many images of White actors and actresses. The average images and pointers to the face sets used are on our OSF page, osf.io/sqm47/.

The Kairos system returns a classification of the sex and race of each face. We used this to assess the effectiveness of the transformations, at least as assessed by this CNN (it is a test of our manipulation, not of the CNN). The majority of the Glasgow face database participants are White; the biggest minority have South Asian origins. Rather than picking 'suitable' faces for transformation, we used the whole set. This was to ensure the faces used were not already biased to match our transforms. Table 1 shows the Kairos system classification for the Glasgow faces before and after each transformation. The CNN clearly indicates that the race transform has mostly worked; the sex transform works for women but less completely for men (some of whom have beards).

# 3. Results and discussion

## 3.1. Human matching tasks

Table 2 shows the performance of the CNNs and human participants on the four matching tasks. It is apparent that the computer systems mostly far out-perform the humans, who average around 70% correct for both match and mismatch trials on these tasks. CNNs 1 and 4 show near-perfect performance, while others would improve with a different decision threshold—the decision boundary between match and mismatch. A high threshold means that a higher similarity score is required for a match to be declared. We used the manufacturer-recommended threshold in all tests. For these images, CNN 3 could afford to be higher, reducing false positives on mismatch trials, and CNN 5 lower, which would improve the hit rate on matches. To compare performance with thresholds that are optimal for these tasks, the area under the curve is also shown. An AUC of 1 is perfect, meaning that it can detect all the true matches for no false matches. CNN 1 made only six errors out of 700

**Table 2.** Performance of the CNNs and humans on the four matching tasks: match/mismatch per cent correct. AUC is area under the curve across all tests combined, calculated using *perfcurve* in Matlab. Human AUC data comes from the average response to each face pair and therefore reflects 'the wisdom of the masses'.

|  | 1 | 2 | 3 | 4 | 5 | 6 | human |
|---|---|---|---|---|---|---|---|
| Kent | 99.5/95 | 83/95 | 99.5/80 | 98.5/95 | 77.5/100 | 83/100 | 77.6/63.8 |
| Models | 100/97.8 | 91.1/95.6 | 100/77.8 | 100/97.8 | 75.6/97.8 | 93.3/97.8 | 67.8/77.2 |
| Make-up | 100/98.6 | 80.2/100 | 100/87.7 | 98.6/99.3 | 79.6/100 | 98.6/87.1 | 73.9/79.8 |
| Dutch | 97.9/100 | 41.7/97.9 | 100/83.3 | 97.9/100 | 35.4/97.9 | 29.2/89.6 | 70.5/74.9 |
| AUC | 0.9997 | 0.9824 | 0.9974 | 0.9979 | 0.9823 | 0.9859 | 0.9671 |

**Table 3.** Percentage of sex and race transformed faces reported as 'matched' to different original images of the same person (figure 1 for illustration).

| transform direction | 1 | 2 | 3 | 4 | 5 | 6 |
|---|---|---|---|---|---|---|
| male -> female | 100 | 93.5 | 100 | 98.8 | 42.9 | 99.4 |
| female -> male | 99.2 | 88.6 | 98.5 | 91.7 | 30.3 | 97 |
| male; White -> Black | 76.4 | 19.4 | 55.8 | 31.1 | 0.5 | 31.8 |
| female; White -> Black | 43.9 | 6.1 | 44.7 | 11.3 | 0 | 18.3 |

trials, including one where it could not find the face. The human score, and this is averaged over many participants, not individual scores, corresponds to about 90% of matches at a false match rate of 10%.

For the human data, there is a slight bias to say mismatch in the last three tests where there are equal numbers of match and mismatch trials. The Kent set has 200 match trials and only 20 mismatches, which is a more realistic ratio in applied contexts. Humans respond by being more likely to say match overall. By default, computer systems have a fixed threshold and are not affected by match–mismatch ratios. It is, however, possible to change the decision threshold depending on the relative likelihood and cost of errors in either direction. Thus, if face recognition is used to permit access to a secure area or bank account, a high threshold might be set to minimize the risk of fraudulent access. Conversely, a low threshold might be used if looking for face matches in a family photograph collection, which would give the best chance of finding people at the minor inconvenience of increasing false matches.

## 3.2. Transformed images

Table 3 shows the results of testing the CNNs on the transformed face tasks. Most systems are largely blind to the sex transform, declaring the transformed faces to be a match. The results from the race transform are more variable, ranging from 0 to 76% of images declared a match. The striking finding is that system 1, that does best in the four standard face-matching tests, makes the most 'errors' (i.e. declares the most matches) on the transformed faces. Table 4 shows the almost perfect Spearman rank correlation between performance on the four face-matching tasks and matches on the transformed images. This correlation appears to be driven more strongly by the match scores, totalled across all four tests, than mismatches.

These findings demonstrate that highly performing current CNNs are relatively blind to face changes that would cause a human observer to reject a match out of hand, on the grounds of being a different apparent sex or race. One possible explanation is that the CNNs use a completely different similarity metric to human observers. We can start to investigate this by looking at the rank correlations between the confidence scores returned by the CNNs and the average accuracy of humans for each trial. High correlations would indicate that the CNNs and humans have a similar ordering of difficulty of matching the pairs, whereas low correlations would suggest different matching strategies.

Table 5 shows the rank correlations, averaged across all four tests via a Z-transform, between each CNN and with the human data. The correlations between the CNNs are high, with an overall average of 0.72 for matches and 0.57 for mismatches. The correlations between the CNNs and the human data are remarkably consistent across systems, averaging 0.41 for match trials and 0.27 for mismatch trials.

**Table 4.** Rank correlations between CNN performance (AUC, matches and mismatches averaged over all four tests) and declared matches on transformed faces (* significant at 0.05, ** at 0.01, two-tailed, N = 6).

|  | AUC | matches | mismatches |
|---|---|---|---|
| sex change | 0.78 | 0.93** | 0.75 |
| race change | 0.83* | 0.89* | 0.67 |

**Table 5.** Average Spearman rank correlations between CNNs and human performance by item, match trials in italics, top right and mismatch bottom left.

|  | 1 | 2 | 3 | 4 | 5 | 6 | human |
|---|---|---|---|---|---|---|---|
| 1 |  | *0.68* | *0.75* | *0.82* | *0.61* | *0.73* | *0.4* |
| 2 | 0.33 |  | *0.71* | *0.77* | *0.67* | *0.79* | *0.46* |
| 3 | 0.48 | 0.58 |  | *0.76* | *0.66* | *0.76* | *0.32* |
| 4 | 0.74 | 0.55 | 0.68 |  | *0.64* | *0.77* | *0.44* |
| 5 | 0.41 | 0.6 | 0.65 | 0.55 |  | *0.65* | *0.38* |
| 6 | 0.44 | 0.67 | 0.59 | 0.64 | 0.57 |  | *0.44* |
| human | 0.27 | 0.22 | 0.28 | 0.3 | 0.27 | 0.29 |  |

It is noteworthy that all of the correlations for matches are higher than all of the correlations for mismatches. The CNNs and humans agree more about what makes faces look similar than they do about what makes them look different. This is consistent with our main finding: there are differences in the appearance of faces which signal a change of identity to humans but not to CNNs.

A correlation of 0.4 is similar to what you might expect for humans doing two different face tasks (e.g. [15]). So, while the CNNs agree more with each other than they do with the humans, the similarity metrics used clearly have something in common. Indeed, it has been reported that a CNN and human observers have a similar hierarchy of the salience of changes to facial appearance. Abudarham *et al.* [16] made controlled changes to faces and found a correlation between the similarity scores given by human observers and the match scores from a CNN.

It is important to note that our data do not speak to the issue of whether CNNs show differential performance between races or sexes (see [17] for a recent survey). What we are reporting is that the networks are relatively blind to variations on these dimensions that humans regard as highly salient. It could be said that CNNs do not share our preconceptions.

That this is so is quite surprising. O'Toole *et al.* [18,19] found that a very simple computational model using principal components analysis of face images identified race and sex as highly salient characteristics. Their models were driven purely by the image properties but the same thing happens in systems that are trained to classify faces by identity. Dahl *et al.* [20] and Kramer *et al.* [21] built simple models that learn identity by using linear discriminant analysis on pre-processed face images and found that sex and race emerge as the first two classification dimensions. CNNs can certainly be trained to identify sex and race; results from the Kairos system are used here as evidence that our transforms had the expected effect. In fact it has been shown that the top-level layer of a CNN that was trained to perform identification codes not only information about sex and race but a range of other social characteristics such as being warm, impulsive or anxious [22].

It would therefore be possible to add a rule that explicitly says, if the images do not match on apparent sex or race, they are not the same person. Doing so, along with other biometric information, such as the person's height, might improve recognition reliability [23]. For example, it has been found that training a CNN explicitly to extract information about the age and sex of a face aids recognition [24]. Whether it would be desirable to make it a rule is a matter for public debate.

## 4. Conclusion

The challenge in face matching is to see past the variations caused by changes in factors such as lighting, viewpoint, expression and age to decide whether two images depict the same identity. CNNs are trained

by showing millions of images, depicting thousands of identities and learning to match pictures to a specific identity. They learn to recognize the variety of images that can represent the same person and to ignore the non-identity variation. It appears that the CNNs that are best able to ignore this extraneous variation are also most prone to ignoring variations that are salient to humans. At one level of description, these nets are doing very well: it really is the same underlying face, transformed in a way that the net has never been exposed to before. The CNNs may therefore declare the face to be a match, which is not necessarily wrong, just surprising from a human perspective.

Ethics. All data collection at the University of Stirling was approved by the General University Ethics Panel and was conducted in accordance with the British Psychological Society guidelines.

Data accessibility. Human and computer performance data are available at https://osf.io/sqm47/.

Authors' contributions. P.J.B.H. conceived the work, ran the computer comparisons, performed analysis and drafted the paper; R.S.S. coded interfaces for computer comparisons and reviewed draft; V.R.M. created the make-up task and collected human data, and reviewed draft.

Competing interests. We declare we have no competing interests.

Funding. P.J.B.H. and V.R.M. were supported by EPSRC grant no. EP/N007743/1, Face Matching for Automatic Identity Retrieval, Recognition, Verification and Management. R.S.S. is supported by the Dylis Crabtree Scholarship.

Acknowledgements. Gill Birtley helped compile the 'Dutch' matching task. Lukas Mayer assisted with data collection. Dan Carragher commented on a draft.

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
