## [Reviewer comments · Royal Society Open Science]

Review History

RSOS-200595.R0 (Original submission)

Review form: Reviewer 1

Is the manuscript scientifically sound in its present form?

Yes

Are the interpretations and conclusions justified by the results?

Yes

Is the language acceptable?

Yes

Do you have any ethical concerns with this paper?

No

Have you any concerns about statistical analyses in this paper?

No

Recommendation?

Accept as is

Comments to the Author(s)

I think your paper is well constructed and helps to build additional evidence that CNNs do not perceive as people perceive. For that reason, I say thank you.

Now let me give a few broad reflections upon how you situate your work in the broader arc of the literature. To start, it simply surprises me to not see a single citation to any of the work by Alice O'toole. Her scholarship surrounding the junction between the psychology of human face recognition and automated face recognition spans over two decades. Just to pick an early work slightly at random "The perception of face gender: The role of stimulus structure in recognition and classification" seems relevant given what you are doing.

More generally your literature review looks to me like it could benefit from a bit more consideration of how people perceive identity and how people perceive gender. Along a similar vein, often discussed is the psychology literature is the difference between expert (familiar) face recognition versus recognition of unknown people. More generally this ties into definitions of expert face recognition - see just for example early work "Training 'greeble' experts: a framework for studying expert object recognition processes". This background is relevant to your work because you seem surprised that CNNs trained not to care about gender do not care when you artificially 'alter' gender appearance. I am not expert myself on the psychology here but postulating separate mechanisms in people for gender and identity is hardly controversial and unless I missed it this issue seemed ignored in your motivation and context.

Last but not least for context - you make no mention of the entire body of face recognition work based upon attributes - including gender. If you are going to suggest gender should be a factor in automated facial recognition two or three sentences giving a nod to this entire line of work seems wise. As a starting point into that literature consider "A.K. Jain, S.C. Dass, K. Nandakumar, Soft biometric traits for personal recognition systems,"

If you spot a theme in my comments it is a request to situate your work more firmly in the history of the scientific discipline. Your experiments are a fine addition and as I already said, not really surprising (to me). Perhaps with a bit more context you can explain a bit more in context rather than simply present your finding.

Last but not least, note I am not asking for changes. Your paper is fine as is. All of my broad comments are simply meant to hopefully spur you on to think a bit further back in the literature and a bit more generally about why perhaps you observe what you observe. I also note you start down this line of reasoning in your conclusion (I like your conclusion) - but again notice your conclusion has no citations. In essence, you find something 'surprising' which to me seems less then surprising - and it is not surprising to me in part based upon what I know broadly of the work that has come before on human face recognition.

Review form: Reviewer 2

Is the manuscript scientifically sound in its present form?

No

Are the interpretations and conclusions justified by the results?

No

Is the language acceptable?

Yes

Do you have any ethical concerns with this paper?

No

Have you any concerns about statistical analyses in this paper?

No

Recommendation?

Major revision is needed (please make suggestions in comments)

Comments to the Author(s)

The paper presents an interesting finding about how CNN works for the task of face recognition. Overall, I see the finding of the paper is interesting, however, I also want to discuss some problems:

- The paper only reports the results based on statistical tests on current face recognition systems. No technical reasons are discussed. For example, why CNNs show this behavior on the face recognition task?
- It is clear that the face transform method used in the paper strongly affects the results. I would like to see how other transform methods (e.g. with noise, etc.) can lead to the same results and conclusion?

Decision letter (RSOS-200595.R0)

Dear Dr Hancock,

The editors assigned to your paper ("Convolutional neural net face recognition works in non-human-like ways") have now received comments from reviewers. We would like you to revise your paper in accordance with the referee and Associate Editor suggestions which can be found below (not including confidential reports to the Editor). Please note this decision does not guarantee eventual acceptance.

Please submit a copy of your revised paper before 26-Jun-2020. Please note that the revision deadline will expire at 00.00am on this date. If we do not hear from you within this time then it will be assumed that the paper has been withdrawn. In exceptional circumstances, extensions may be possible if agreed with the Editorial Office in advance. We do not allow multiple rounds of revision so we urge you to make every effort to fully address all of the comments at this stage. If deemed necessary by the Editors, your manuscript will be sent back to one or more of the original reviewers for assessment. If the original reviewers are not available, we may invite new reviewers.

When submitting your revised manuscript, you must respond to the comments made by the referees and upload a file "Response to Referees" in "Section 6 - File Upload". Please use this to document how you have responded to the comments, and the adjustments you have made. In

order to expedite the processing of the revised manuscript, please be as specific as possible in your response.

- Data accessibility

If you wish to submit your supporting data or code to Dryad (<http://datadryad.org/>), or modify your current submission to dryad, please use the following link:
<http://datadryad.org/submit?journalID=RSOS&manu=RSOS-200595>

- Competing interests

- Authors' contributions

- Acknowledgements

- Funding statement

on behalf of Marta Kwiatkowska (Subject Editor)
openscience@royalsociety.org

Associate Editor's comments:

Comments to the Author:

Thank you for the manuscript. The reviewers appear to be broadly positive in relation to your manuscript; however, the second referee has a number of suggestions we would like you to address, and, while the opportunity presents, you might like to examine whether tackling the comments from the first reviewer will add value to the current manuscript (as well as future work, as indicated). Thanks again.

Reviewers' Comments to Author:

Reviewer: 1

Comments to the Author(s)

I think your paper is well constructed and helps to build additional evidence that CNNs do not perceive as people perceive. For that reason, I say thank you.

Now let me give a few broad reflections upon how you situate your work in the broader arc of the literature. To start, it simply surprises me to not see a single citation to any of the work by Alice O'toole. Her scholarship surrounding the junction between the psychology of human face recognition and automated face recognition spans over two decades. Just to pick an early work slightly at random "The perception of face gender: The role of stimulus structure in recognition and classification" seems relevant given what you are doing.

More generally your literature review looks to me like it could benefit from a bit more consideration of how people perceive identity and how people perceive gender. Along a similar vein, often discussed is the psychology literature is the difference between expert (familiar) face recognition versus recognition of unknown people. More generally this ties into definitions of expert face recognition - see just for example early work "Training 'greeble' experts: a framework for studying expert object recognition processes". This background is relevant to your work because you seem surprised that CNNs trained not to care about gender do not care when you artificially 'alter' gender appearance. I am not expert myself on the psychology here but postulating separate mechanisms in people for gender and identity is hardly controversial and unless I missed it this issue seemed ignored in your motivation and context.

Last but not least for context - you make no mention of the entire body of face recognition work based upon attributes - including gender. If you are going to suggest gender should be a factor in automated facial recognition two or three sentences giving a nod to this entire line of work seems wise. As a starting point into that literature consider "A.K. Jain, S.C. Dass, K. Nandakumar, Soft biometric traits for personal recognition systems,"

If you spot a theme in my comments it is a request to situate your work more firmly in the history of the scientific discipline. Your experiments are a fine addition and as I already said, not really surprising (to me). Perhaps with a bit more context you can explain a bit more in context rather than simply present your finding.

Last but not least, note I am not asking for changes. Your paper is fine as is. All of my broad comments are simply meant to hopefully spur you on to think a bit further back in the literature and a bit more generally about why perhaps you observe what you observe. I also note you start down this line of reasoning in your conclusion (I like your conclusion) - but again notice your conclusion has no citations. In essence, you find something 'surprising' which to me seems less then surprising - and it is not surprising to me in part based upon what I know broadly of the work that has come before on human face recognition.

Reviewer: 2

Comments to the Author(s)

The paper presents an interesting finding about how CNN works for the task of face recognition. Overall, I see the finding of the paper is interesting, however, I also want to discuss some problems:

- The paper only reports the results based on statistical tests on current face recognition systems. No technical reasons are discussed. For example, why CNNs show this behavior on the face recognition task?

- It is clear that the face transform method used in the paper strongly affects the results. I would like to see how other transform methods (e.g. with noise, etc.) can lead to the same results and conclusion?

Author's Response to Decision Letter for (RSOS-200595.R0)

See Appendix A.

RSOS-200595.R1 (Revision)

Review form: Reviewer 2

Is the manuscript scientifically sound in its present form?

Yes

Are the interpretations and conclusions justified by the results?

Yes

Is the language acceptable?

Yes

Do you have any ethical concerns with this paper?

No

Have you any concerns about statistical analyses in this paper?

No

Recommendation?

Accept as is

Comments to the Author(s)

The authors did a good job to revise the paper. Although there are some technical aspects that I want to discuss more, but generally the paper is in good form and in my opinion can be accepted.

Decision letter (RSOS-200595.R1)

Dear Dr Hancock,

It is a pleasure to accept your manuscript entitled "Convolutional neural net face recognition works in non-human-like ways" in its current form for publication in Royal Society Open Science. The comments of the reviewer(s) who reviewed your manuscript are included at the foot of this letter.

Kind regards,

Andrew Dunn

on behalf of Prof Marta Kwiatkowska (Subject Editor)

Reviewer comments to Author:

Reviewer: 2

Comments to the Author(s)

The authors did a good job to revise the paper. Although there are some technical aspects that I want to discuss more, but generally the paper is in good form and in my opinion can be accepted.

Appendix A

**UNIVERSITY of
STIRLING**

Psychology
Faculty of Natural Sciences
University of Stirling
Stirling
FK9 4LA
Scotland UK

T : +44 (0) 1786 467675
E : pjbh1@stir.ac.uk
W : www.psychology.stir.ac.uk

25 June 2020

Dear Anita Kristiansen ,

We thank the action editor and referees for their comments on our paper and hope the revisions we have made will be acceptable. We have left the amendments made highlighted in the text, to make it easy to see what has changed.

The first referee made two suggestions for additional background material, which we have addressed with added paragraphs. We are certainly happy to acknowledge the body of work led by Alice O'Toole, though we believe that some of her other work is more relevant than the paper suggested. We have included some of this in added paragraphs in the introduction, supporting the idea that people do regard perceived sex and race as salient characteristics.

We have also added some more text and a couple more references to the discussion about the possibility of training networks to extract race and sex information explicitly.

The second referee asked two interesting questions. The answer to both is well beyond the scope of this revision; understanding why CNNs behave as they do is the subject of many research projects. Hopefully this paper will add to their enquiries.

We are very aware of the potential sensitivities of the paper, should it be accepted. In fact, Anna Bobak, whose contribution to the work was the smallest, has asked to be removed from the list of authors due to these concerns. We have pondered how to reduce the potential for misunderstanding and offence and have added a paragraph about the transforms we are using. We have added a word such as 'apparent' whenever we talk about race transforms. We have also changed any references to African American to the more usual UK usage of Black. Following the advice of the American Psychological Society, we have used capitals for White and Black. We would very much welcome any advice that editors and staff at the journal might offer about these issues.

If the paper is accepted, we plan to do a press release and possibly a Conversation article. It would be as well to coordinate our publicity with any that the journal might make.

During the upload process, I noticed that Dr Viktoria Mileva's prefix is incorrectly given as Professor. Her Orcid id is not given; it is [0000-0002-7983-3069](https://orcid.org/0000-0002-7983-3069). Neither of these fields is editable from that interface.

Yours faithfully,

Peter Hancock